# Determining Optimal Temperature Combination for Effective Pretreatment and Anaerobic Digestion of Corn Stalk

**DOI:** 10.3390/ijerph19138027

**Published:** 2022-06-30

**Authors:** Juan Li, Xiujin Li, Akiber Chufo Wachemo, Weiwei Chen, Xiaoyu Zuo

**Affiliations:** 1Department of Environmental Science and Engineering, Beijing University of Chemical Technology, No. 15 Beisanhuan East Road, Chaoyang District, Beijing 100029, China; lijuan19900801@163.com (J.L.); xjli@mail.buct.edu.cn (X.L.); akiberchufo@yahoo.com (A.C.W.); 13311558416@163.com (W.C.); 2Faculty of Water Supply and Environmental Engineering, Arba Minch University, Arba Minch P.O. Box 21, Ethiopia

**Keywords:** anaerobic digestion, pretreatment, NaOH, corn stalk, mesophilic, thermophilic

## Abstract

Temperature is one of the important factors affecting both chemical pretreatment and anaerobic digestion (AD) process of corn stalk (CS). In this work, the combined ways between pretreatment temperature (40 °C and 60 °C) and AD temperature (35 °C and 55 °C) were selected to investigate the AD performance for sodium hydroxide (NaOH) pretreated CS. Three organic loading rates (OLRs) of 1.6, 1.8 and 2.0 g·L^−1^·d^−1^ were studied within 255 days using continuously stirred tank reactors (CSTR). The results revealed that biogas yields of CS after pretreated were higher than that of untreated groups by 36.79–55.93% and 11.49–32.35%, respectively. When the temperature of NaOH pretreatment changed from 40 °C to 60 °C, there was no significant difference in enhancing the methane yields during the three OLRs. The mesophilic AD (MAD) of CS pretreated with 2% NaOH under 40 °C and 60 °C conditions produced 275 and 280 mL·g_vs_^−1^ methane yield at OLR of 1.6 g·L^−1^·d^−1^. However, as the OLR increased, the methane yield of CS under thermophilic AD (TAD) condition was further higher than under MAD condition. Furthermore, from the perspectives of energy balance and economic analysis, AD of 40 °C-treated CS recovered more energy and TAD is less expensive. Therefore, temperature of 40 °C was considered as an appropriate for pretreatment whether in mesophilic or thermophilic AD system. On the other hand, TAD was chosen as the optimal AD temperatures for higher OLRs.

## 1. Introduction

Production of biogas from corn stalk (CS) has been growing in different parts of the world and is considered as ecofriendly energy source. Due to the high abundance and carbohydrate content, the bioconversion of CS using anaerobic digestion (AD) technology has great potential for biogas production [1]. The main components of CS are cellulose, hemicellulose, and lignin (LCH). Cellulose chains with many hydroxylic groups are linked with hydrogen bonds, resulting in high tensile strength. Meanwhile, cellulose molecules have complicated polymer crystal structures owing to their different orientations. Lignin is a polymer of phenylpropane units, which crosslinks with cellulose and hemicellulose and forms a three-dimensional network inside the cell wall [2]. Due to the high polymerisation, heterogeneity, crystallinity of lignocellulose fibres, it is difficult for anaerobic microorganisms to use it directly [3]. Therefore, pretreatment of lignocellulose to change lignin structure, reduce cellulose crystallinity and increase surface area was indispensable for improving biodegradability and biogas production in AD process [4,5].

Nowadays, there were a number of methods available for the pretreatment of CS including physical [6,7], chemical [8,9], biological [10,11] and combined pretreatments [12]. Touching up these methods, physical pretreatment is among the most basic that can be used to pretreat lignocellulosic substrates, involving comminution, steam-explosion, liquid hot water, extrusion, heating, and irradiation, etc. [13,14,15,16]. Chemical pretreatment including alkali and acid pretreatments, which currently is the most studied method for treating lignocellulose [17]. Among different alkali chemical pretreatment methods, ammonia solution, urea, NaOH, KOH, Ca(OH)_2_ and deep eutectic solvent (DES) are commonly used chemical reagents. Ammonia solution and urea pretreatments can cause ammonia inhibition in the AD reactor system, which will thus reduce biogas production [18,19]. From an economic point of view, the KOH reagent is actually too expensive to be suitable for pretreatment. The actual pretreatment effect of Ca(OH)_2_ is not suitable for CS and it is difficult to destroy the lignocellulosic structure [20]. The deep eutectic solvent (DES), as a new synthetic green solvent, has the advantages of low-cost effectiveness, non-volatility and good thermal stability [21,22,23]. However, some DES have the ability to inhibit bacteria, fungi and viruses due to the different composition of DES (hydrogen bond acceptor and hydrogen bond donor) and the chemical structure of the mixture [24]. NaOH as one of the most effective alkali [25,26], which is characterized by mild conditions, high lignin removal rate and high biogas yield, has been widely used in pretreatment process [27,28,29]. For example, Li et al. reported that 2% NaOH pretreated CS at solid state under 20 °C condition achieved methane production of 211 mL·g_vs_^−1^ for mesophilic AD (MAD) [30]. Zheng et al. also found that 2% dose of NaOH-treated CS at ambient temperature (20 °C) produced optimum 233 mL·g_vs_^−1^ methane at mesophilic anaerobic digestion (MAD) condition [31]. While using NaOH regents to break the complex structures of lignocellulosic compounds, pretreatment temperature is one of the decisive factors [32]. However, these research mainly focused on normal temperature pretreatment and operated on MAD condition, there is a lack of studies on high temperature pretreatment and thermophilic AD (TAD).

Moreover, the microorganisms in anaerobic reactors are highly sensitive to the changes in temperature [33]. High thermophilic AD temperature can improve hydrolysis and physical degradation of the substrate, increases destruction of organic matter and enhances methane yields during anaerobic process [34]. Due to the natures of microorganisms which can facilitate anaerobic processes in reactors and bioconversions which are commonly carried out under either mesophilic (30–45 °C) or thermophilic (50–65 °C) conditions. Among the biochemical stages in anaerobic process, methanogenesis performs effectively at 35 °C and 55 °C reactor temperatures [35]. When the AD temperature increases to 55 °C, the rate of methane production and yield from substrates such as cellulose, food waste, and maize silage will increase [36]. More specifically, using TAD for biomethane production has some advantages over MAD; such as higher metabolic and methane production rates, shortened hydraulic retention time, and higher degree of pathogen reduction [37,38]. In addition, Watanabe et al. reported that due to the presence of optimum rate of lignocellulose degradation, the process of TAD produced high methane yield of 220 mL·L^−1^·d^−1^ from CS [39].

A number of digesters are being applied for AD process in both laboratory study and industrial project, completely stirred tank reactor (CSTR) is one of most commonly used for the AD of high-solid wastes such as straws. The digestate of CSTR discharged each day actually is a mixture of well digested and partially digested due to different times of feeding and staying in the digester [36]. In real-world applications, CS needs to be reacted all year round to produce biogas, so the reactor is also operated all year round, and for such a reactor, the pretreatment and AD temperatures of raw materials are very important. Moreover, there are few studies on the long-term operation of CS under TAD condition.

Therefore, the main objective of this study was to evaluate long-term effect of biogas and methane production of CS under different pretreatment and AD temperature conditions, and to determine the optimal combination measure of pretreatment and AD. Different pretreatment temperatures were used to evaluate AD performance of CS in both mesophilic and thermophilic conditions. Furthermore, the components conversion rate and system stability during entire MAD and TAD periods were analysed. The energy balance was also considered to evaluate comprehensive AD process.

## 2. Materials and Methods

### 2.1. Feedstock and Inoculum

The CS used in this study was collected from farmland at Beijing suburb of Shunyi district, China. The collected CS was dried at open air until it achieves a moisture content of less than 10% and then stored in cool and dry place in an incubator at room temperature until the next use. Then the dried CS was ground to the size of 20-mesh using a lab mill (YSW-180, Zhengde, China). The inoculum was collected from stable biogas plant continuously operated under mesophilic condition at Shunyi District, Beijing, China. The obtained inoculum was taken and incubated at 35 °C and 55 °C temperature in our laboratory for 10 days. The main components and characteristics of CS before AD and seeding sludge before incubation are listed in Table 1.

### 2.2. Pretreatment with NaOH

In order to improve the biodegradability, the grounded CS powder was pretreated with 2% NaOH (2 g/100 gTS of CS) sodium hydroxide solution based on the ratio of dry weight of CS:H_2_O:NaOH = 1:6:0.02 (*w*/*w*/*w*) at two different temperature (40 °C and 60 °C) for two days in constant temperature incubators. Then the pretreated CS was used for AD tests. This work was operated under mesophilic temperature (35 °C) and thermophilic temperature (55 °C) using continuously stirred-tank reactors (CSTRs).

### 2.3. Experimental Set-Up

In this study, six CSTRs (LanYu Environment Co., Ltd., Chuzhou, Anhui, China) with a working volume of 8 L and free headspace of 2 L were employed for AD tests. Three mesophilic CSTRs were made from transparent polymethyl methacrylate plastic and operated at temperature of 35 °C, and the other three thermophilic CSTRs were made from stainless steel and operated at temperature of 55 °C. The reactors were equipped with inlet and outlet valves for feeding and effluent flow, respectively. Each reactor was designed with thermostatically controlled water jacket to maintain the mesophilic and thermophilic temperature. During the digestion process, the agitation of the substrate in the reactor was operated by a top-mounted motor with the stirring rate of 80 rpm at the interval of 2 h and lasting for 5 min for each stirring [40]. The daily biogas production was monitored by wet gas meter (LMF-1, Changchun Automobile Filter, Changchun, China).

### 2.4. Operating Conditions

The six CSTRs undergone through start-up process, when hydraulic retention time (HRT) continued the amount of organic loading rates (OLR) increased during the processes. In this study, each CSTR was started with seeding sludge untreated and NaOH-pretreated CS based on the design. The initial start-up stage concentration of CS loaded in each reactor was 65 gTS·L^−1^ and 15 gTS·L^−1^ of sludge was seeded in each digester following the results of previous research work [41]. The HRT for each OLR was 50 days. Sample CS pretreated at 40 °C was fed into two CSTRs operating in mesophilic and thermophilic conditions. In the same way at 60 °C pretreated CS were also fed into one mesophilic CSTR and the other in thermophilic CSTR. The untreated CS fed into mesophilic and thermophilic CSTRs was used as control group.

After 30 days of start-up period, when the variation in daily biogas production was within 10% of the average value after feeding of everyday in each HRT, the reactors were thought been in stabilization conditions [42]. Then the pretreated CS were fed after every 24 ± 1 h with OLR of 1.6 g·L^−1^·d^−1^ in first round (R1), 1.8 g·L^−1^·d^−1^ in second round (R2) and 2.0 g·L^−1^·d^−1^ in third round (R3), and R1–R3 were ran for 1.5 HRT (75 days) of stable running time.

### 2.5. Analytical Methods

The biogas composition (H_2_, CH_4_, CO_2_, and N_2_) was determined using a gas chromatograph (GC) (SP2100, BeiFenRuiLi, Beijing, China) equipped with a molecular sieve (TDX-01) packed in 2 m × 3 mm stainless-steel column and a thermal conductivity detector (TCD). The temperatures of oven, injector port, and TCD were 140, 150, and 150 °C, respectively. Argon was used as the carrier gas where its flow rate was 30 mL·min^−1^. Daily methane production was determined from CH_4_ content, and the amount of biogas produced was reported after converting to standard temperature and pressure conditions. Biogas and methane production value were calculated and presented based on the amount of VS fed.

TS, VS and MLSS of CS, inoculum and their mixture were determined according to the APHA standard methods [43]. The total carbon (TC) and total nitrogen (TN) were determined by the Vario EL/micro cube elemental analyzer (Elementar, Hanau, Germany). The pH was directly measured with a pH meter (3-Star, Thermo Orion, Waltham, MA, USA). Alkalinity was determined by pH titration method using 0.1 M HCl and expressed in g equivalent CaCO_3_·L^−1^ using the APHA standard methods. The composition of cellulose, hemicelluloses and lignin in CS and digestate were measured using the Fiber Analyzer (ANKOM, A2000i, New York, NY, USA). Ammonia nitrogen concentration was measured by Kjeldahl analyzer (KT-260, Foss, Denmark).

The samples for volatile fatty acid (VFA) and ethanol analysis were initially centrifuged at 10,000 rpm for 10 min, the supernatants were then filtered through a 0.22 μm filter, and finally the filtrates were collected in sample vials for analyses. The VFAs values were calculated as sum of the measured acetic (HAc), propionic (HPr), n-butyric (n-HBu), iso-butyric (iso-HBu), n-valeric (n-HVa) and iso-valeric (iso-HVa) acids. The content of each VFA was determined using the gas chromatograph (GC2014, Shimadzu, Japan) equipped with a flame ionization detector (FID) and a DB-WAX123-7032 capillary column. Nitrogen was used as the carrier gas. The operational temperatures of injector, detector and column were kept at 250 °C, 250 °C and increased from 100 to 180 °C at a rate of 5 °C·min^−1^, respectively.

All the results were analysed by MS Excel 2013 and figures in this paper were drawn using OriginPro 2019 (OriginLab, Northampton, MA, USA).

### 2.6. Assessment of Energy Balances

In order to assess the energy balances of the AD, the parameters for the reactors were estimated from experimental data. It was assumed that the performances of the reactors were the same as those in the test. Three relative processes were considered: pretreatment, pumping and mixing. The energy input (E_i_) for AD includes the electricity demand for heating the substrate to pretreatment temperature (E_i,h_), pumping energy for heating the substrate to digestion temperature (E_i,p_) and mixing energy for compensating for the heat losses (E_i,m_). The energy output (E_o_) from the AD process was calculated from the methane yield. The calculations were carried out in the same manner as previous studies [44,45]. The ratio of E_o_ to E_i_ (R_o/i_) represents another target to evaluate the energy gains or losses. Energy balance (ΔE and R_o/i_) was calculated based on Equations (1) and (2) as follows:E = E_o_ − E_i_ = E_o_ − (E_i,h_ + E_i,m_ + E_i,p_)(1)
R_o/i_ = E_o_/E_i_(2)

## 3. Results and Discussion

### 3.1. Daily Biogas Production and Average Methane Content

The daily biogas production (DBP) of MAD and TAD digesters of 40 °C and 60 °C pretreatment condition are shown in Figure 1a,b. During starting up period, two biogas production peaks appeared, with the longer of the AD time, the daily biogas production slowly approached zero. This is because the substrate is continuously consumed and degraded by microorganisms, and finally rarely decomposable products in reactor. The results from Figure 1 revealed that in MAD system (Figure 1a), the DBP of CS dropped slowly from 6.25 L·d^−1^ to 5.62 L·d^−1^ as the OLR increased from R1 to R3 of two treatment conditions while untreated one was lower than that and showed no obvious increment. As shown in Table 2, during MAD processes, the average methane content showed increment from 52.49% to 56.90% at 40 °C condition and from 53.05% to 57.52% at 60 °C condition. However, in TAD system (Figure 1b), the DBP of CS gradually improved from 6.12 L·d^−1^ to 7.63 L·d^−1^ as the OLR increased from R1 to R3 of two treatment conditions while untreated one showed a little improvement. During this TAD processes, the average methane content, respectively, kept stable at 52.48% and 52.09% of 40 °C and 60 °C pretreatment condition for all OLRs as shown in Table 2.

The relatively fast DBP rates of CS in TAD systems was due to the thermophilic hydrolytic bacterial population which would accelerate the hydrolysis process, mainly at a temperature of around 55 °C [46]. In addition, high rate of hydrolysis increases the production of volatile fatty acids (VFAs) from CS is also more efficient at TAD reactors than at mesophilic digestion [47]. Therefore, the daily biogas productions at TAD conditions were higher than MAD conditions.

### 3.2. Influence of OLR on Reactor Performance

A change from low OLR to high OLR may lead to system instability, such as accumulation of VFAs, a lower pH, and a mismatch between the growth rates of VFA-producing and consuming microbes and thus resulted in low biogas production [48,49]. This was consistent with the results of Figure 1a, which demonstrated the production rates of biogas decreased in MAD systems as OLR increased. Nevertheless, the production rates of biogas improved in TAD systems as OLR increased. At the same time, in TAD system, the feeding rate of 2.0 g·L^−1^·d^−1^ was 1.25 times of 1.6 g·L^−1^·d^−1^, which produced enhanced biogas production rate by 24.59% and 38.18% than that of MAD and untreated conditions, respectively. This was because the consuming speeding of soluble substance in TAD system higher than in MAD systems. As a result, the TAD systems achieved higher biogas production rate relative than MAD and untreated systems as OLR increased. Furthermore, the OLR (1.6 g·L^−1^·d^−1^) of feeding in this study were higher than that of another study of Tian (1.44 g·L^−1^·d^−1^) [50], and the DBP in this study were also higher than that study, which only attained 4.83–5.30 L^−1^·d^−1^. This illustrated that the OLRs included in this study were appropriate for AD digesters.

### 3.3. Comparison of Different Pretreatment and AD Temperatures

#### 3.3.1. Comparison of Different Pretreatment Conditions

Pretreatment temperatures of biomass materials usually play an important role in batch experiment but may not intrigue decisive effects of CSTR study [51]. In this work, it is obviously that retreating of CS with NaOH at temperature of 40 °C and 60 °C produced higher methane yield than the control of untreated group. At 40 °C pretreatment condition, the rates of biogas production under MAD were 6.25 L·d^−1^, 5.54 L·d^−1^ and 5.62 L·d^−1^ for three respective OLRs. This result was 36.81–38.69%higher than that of untreated, indicating that sodium hydroxide pretreatment can destroy straw structure, thereby increasing biogas production At 60 °C pretreatment condition, the rates of biogas production under MAD were 6.24 L·d^−1^, 5.57 L·d^−1^ and 5.60 L·d^−1^ for three respective OLRs. The same trends also presented in TAD system, and likewise, the methane production of different pretreatment conditions indicated similar results as well. As shown in Table 2, the gaps of methane yield between 40 °C and 60 °C pretreatment condition were small under MAD conditions which were 5, 9 and 3 mL·g_vs_^−1^ for three OLRs, respectively. These also showed the same trends of the gaps under TAD which were 12, 8 and 0 mL·g_vs_^−1^ for three OLRs, respectively. This could be directly related to the gap of production of TVFAs hydrolyzed from CS between 40 °C and 60 °C. As shown in Figure 2 and Table 3, the TVFAs and ethanol produced under 40 °C and 60 °C pretreatment showed no obvious difference, even the production of acetic acid under 40 °C was a little bit more than that under 60 °C, which not harshly inhibit the production of biogas. In addition, biogas and biomethane productions under all the above conditions is higher than untreated group.

Moreover, the productions of methane in this research were higher by 32.7% than previous study which gained 211 mL g_vs_^−1^ at MAD using CS pretreated in 20 °C [52]. The results were also higher by 15.7% than the study of Liu et al., which gained 242 mL·g_vs_^−1^ under TAD with 20 °C treated CS [33]. These results revealed that under MAD or TAD condition, enhancing the pretreatment temperature from 40 °C to 60 °C was not highly conducive to improve the biogas and methane yield. Based on cost benefit and effectiveness, 40 °C pretreatment temperature could be suggested as the optimum pretreatment temperature for MAD and TAD processes.

#### 3.3.2. Comparison of Different AD Conditions

The process performance of anaerobic reactors often depend on temperature, which plays its part in making favorable conditions for degradable substance in reactors especially in long-term operating condition [53]. Therefore, the Figure 3 and Table 2 showed significant daily methane production (DMP) and methane yields difference between MAD and TAD systems. Firstly, it was also obviously that pretreating of CS with NaOH at AD temperature of 35 °C and 55 °C produced higher methane yield than the control. Then, with the OLR increased, the gap in methane yield between the two AD temperatures was gradually becoming greater. This was because excess VFAs accumulated in reactors inhibited MAD processes but stimulated TAD processes to result different methane yields. In R3 phase (highest OLR phase), the methane yield of TAD was enhanced by 30.38%, 28.26% and 52.54% than that of MAD with 40 °C, 60 °C treatment and untreated one. The methane yield of TAD under 40 °C and 60 °C pretreatment conditions were higher than that by 31.11% compared with untreated CS.

The higher methane yield in TAD system was attributed to the high rate of methanogenesis under thermophilic condition, so that VFA could be easily converted to methane after hydrolysis and acidogenesis process in the thermophilic reactors [54]. Jiménez et al. also reported that specific methane activity (SMA) of thermophilic methanogens was higher than mesophilic methanogens [55]. In a study using wheat straw as raw material, methane yield under thermophilic conditions was higher than that under mesophilic conditions, where the results are in line with the present work [56]. Consequently, the results revealed that thermophilic system can tolerate high OLR feeding and could be an important insight for the future biogas industry using CS as feedstock.

### 3.4. Changes of Main Compositions

Content of LCH after pretreatment was shown in Table 4. Prior to treatment the main components of CS, lignin, cellulose and hemicellulose, accounted for 40.36%, 20.07% and 11.41%, respectively, giving 71.84% in total. Following 40 °C and 60 °C treatments, the proportion of hemicellulose significantly decreased to 13.15%, which was 52.62% lower than that of untreated control. This was because after pretreatment, part of released cellulose and hemicellulose were converted into acid-soluble substances, as shown in Figure 2, which therefore led to the content of cellulose and hemicellulose in pretreated CS are lower than that of untreated [25,32,57]. The content of lignin also decreased by 5.6–7.4%, which suggested partial lignin removed via NaOH pretreatment. Lignin degradation could release more cellulose and hemicellulose, thus increasing the biodegradability of the CS.

LCH were the main carbon sources for anaerobic microorganisms during AD of CS. The biogas and methane production were attributed to biological degradation of cellulose and hemicellulose. The conversion rates of cellulose, hemicellulose, TS and VS during the process of AD were used to evaluate the biodegradability improvement and digestion performance of the system [58]. Conversions rate of cellulose, hemicellulose, TS and VS after AD were analyzed from the effluent discharged after steady-state. As shown in Table 5, compared to the untreated, the reductions of cellulose, hemicellulose and lignin after AD with 40 °C and 60 °C pretreatment improved by 12.63–35.37%, 8.13–32.59% and 26.92–67.95%, respectively. In addition, the TS and VS removal rate of 40 °C and 60 °C pretreatment amounted to 46.3–50.6%and 53.55–61.9% while the untreated one kept 43.3–47.5% and 53.5–58.7%. At the same time, the reductions of all components in TAD systems showed relative higher than in MAD systems. These results were also in agreement with the fact that under thermophilic digestion the rate of organic matter removal is more efficient [59]. The main reason for the improvement of conversion rate at TAD might be the biodegradation of more feedstock which accounts for more methane produced from excess VFAs that released directly from the thermophilic reactor. The high conversion of substrates was also corresponded to the biogas and methane production which presented a better conversion rate of CS under TAD condition from R1 to R3.

### 3.5. Evaluation of System Stability

During the AD processes, the activities of microbial communities can be deeply influenced by pH, ammonia nitrogen (AN) and total volatile fatty acids (TVFAs). More specifically, the loss of methanogens, high accumulation of TVFAs and low total alkalinity concentration (TAC) can inhibit the process of AD. Additionally, the ratio of TVFAs to TAC (TVFAs/TAC) can be used as significant indicators of the stability of AD [60].

#### 3.5.1. pH

The value of pH is an important parameter in determining the stability of the AD system. In this study, all the pH values during entire AD period were in the range of 6.8–7.7 (Figure 4a). This shows that the pH values were at optimum level, which was believed to be suitable for the growth of methanogens [35]. From two AD conditions applied in this test the TAD systems had a relatively higher pH than the MAD systems during all the OLRs. This was mainly due to high consumption rate of VFAs produced at optimum pH under thermophilic condition.

#### 3.5.2. Ammonia Nitrogen and TVFAs/TAC

Optimal ammonia concentration ensures sufficient buffer capacity of methanogenic medium in AD thus increasing the stability of the digestion process. However, high ammonia is regularly reported as the primary cause of digester failure because of its direct inhibition of microbial activity [61]. As shown in Figure 4b, The AN of the process effluent of six CSTRs was high after starting feeding but gradually reduced to 250 mg·L^−1^ at the end of the first OLR (R1) and less than 200 mg·L^−1^ in the second to third OLRs (R2 and R3), which didn’t inhibit the process [62]. At the same time, there were two small AN peak values by the end of later two OLRs for more substrates were degraded by microorganism.

In this study, TVFAs from the effluent of six CSTRs at the OLRs of 1.6, 1.8 and 2.0 g·L^−1^·d^−1^ were determined. As shown in Figure 5a, with the prolongation of AD time, the TVFAs concentration accumulated in the six reactors became lower and lower until the end of each OLR. This was attributed to the number of methanogens increases with the prolongation of time, and the VFAs produced in reactors gradually degraded into biomethane. It also can be seen that the time of TVFAs accumulation of 40 °C pretreated samples were 10 days faster and had more TVFAs than the samples pretreated at 60 °C temperature in the second OLR (R2 stage). In addition, in R3 phase, the TVFAs reached its peak level for reactors with the samples at 40 °C and 60 °C pretreatment temperature both at the 200th day, this was because the gap of amount of TVFAs was small in two treatment conditions. Moreover, the TVFAs produced in TAD were relatively less than that of MAD during R1 to R3 processes while the pH value in TAD were relatively higher than that of MAD, which showed the consistency of TVFAs and pH value.

The ratio of TVFAs/TAC can be applied to evaluate the buffer ability of anaerobic digestion. Generally, the digestion process could keep stable when the ratio of TVFAs/TAC is below 0.4, and however, the system shows instability when the ratio exceeds 0.8 [53]. TVFAs and TVFAs/TAC of this work were shown in Figure 5b. The results showed that the ratios of VFA to alkalinity ration of the effluents from six reactors were from 0.01 to 0.2. This showed that the operating condition in each CSTR was at optimum level. It can be seen that the TVFAs/TAC of untreated group in both AD temperature systems during the three OLRs showed relatively high. This demonstrated that pretreated CS can more easily maintain the system stability during the AD process. Furthermore, the TVFAs/TAC of TAD was lower than MAD during the last two OLRs, which indicated that TAD showed better buffer ability at high OLR.

### 3.6. Energy Balance

As AD is an energy production system, it is important to evaluate the energy balances of the system for the characteristics of the six systems digestates were different (Figure 4). In order to evaluate comprehensively about six AD reactors, the energy input of the systems should also be considered. In this paper, three processes (pretreatment, pumping and mixing) were considered (Equation (1)) and the results are summarized in Table 6.

The input energy includes the electricity and heat input [63]. The input electricity of the TAD systems was slightly higher than those of the MAD systems because the power and heating of TAD reactors have more demands. The input heat of the 40 °C-treatment was slightly lower than that of 60 °C-treatment and E_i,h_ of the two temperatures were the same under different AD conditions. The energy balance analysis of AD alone (Table 6) showed the ΔE of the three MAD systems were higher than that of TAD systems, which was consistent with the value of R_o/i_ of six different reactors. The results suggested the MAD of CS in this study recovered more energy from the degraded organic matter of CS than the thermophilic digestion although it produced less energy from methane gas. In addition, the ΔE of 40 °C-treatment CS were relatively higher than that of 60 °C-treatment while R_o/i_ also showed high, this demonstrated the AD of 40 °C-treated CS recovered more energy. As a result, 40 °C was recommended as a preferable pretreatment temperature.

### 3.7. Economic Aspect of Pretreatment and Anaerobic Digestion

Evaluation of economic analysis (Table 7) for one ton of CS with untreated and two pretreatment conditions was provided by engineering economics analysis method was used in this study according to the market prices in China. The final result of Renminbi (CNY) would be converted into US dollars (USD) based on the recent exchange rate (CNY/USD = 0.1494). The highest OLR (2.0 g_vs_·L^−1^·d^−1^) was taken as an example for the methane yield, and then the methane gas produced in untreated groups achieved highest unit methane production cost, which was 12.95–48.93% higher than that of other groups. It can be seen from Table 7 that cost of 40 °C and 60 °C pretreated CS were 0.0833 USD·m^3^·CH_4_^−1^ and 0.0867 USD·m^3^·CH_4_^−1^, respectively. The cost under different pretreatment conditions is basically the same, however, the cost of TAD conditions is lower than that of MAD conditions. Therefore, the TAD process of CS has advantage of cost effective as compared to MAD. For the reason that TAD was recommended as a preferable anaerobic fermentation condition especially with high OLRs.

## 4. Conclusions

This study investigates the AD performance for NaOH pretreated CS with the combined ways between two pretreatment temperatures (40 °C and 60 °C) and AD temperatures (35 °C and 55 °C). Three OLRs of 1.6, 1.8 and 2.0 g·L^−1^·d^−1^ were studied within 255 days using continuously stirred tank reactors (CSTR). The results revealed that biogas yields of CS after pretreated were higher than that of untreated groups by 36.79–55.93% and 11.49–32.35%, respectively. The MAD of CS pretreated with 2% NaOH under 40 °C and 60 °C conditions produced 275 and 280 mL·g_vs_^−1^ methane yield at OLR of 1.6 g·L^−1^·d^−1^. When the temperature of NaOH pretreatment changed from 40 °C to 60 °C, there was no significant difference in enhancing the methane yields during the three OLRs. As the OLR increased, the methane yield of CS under TAD condition was further higher than under MAD condition. Therefore, the optimum pretreatment temperature of CS was 40 °C for both MAD and TAD conditions, and the energy balance and economic analysis also supported this result. At the same time, TAD at high OLR and MAD at low OLR were suggested as the optimal and economically better for the methane production from CS. Moreover, the TAD groups achieved lowest cost as compared to untreated and MAD groups. The results also revealed that thermophilic system can tolerate higher OLR feeding way and TAD could be considered as an important way to produce biomethane when high OLR demanded in engineering.

## Figures and Tables

**Figure 1 ijerph-19-08027-f001:**
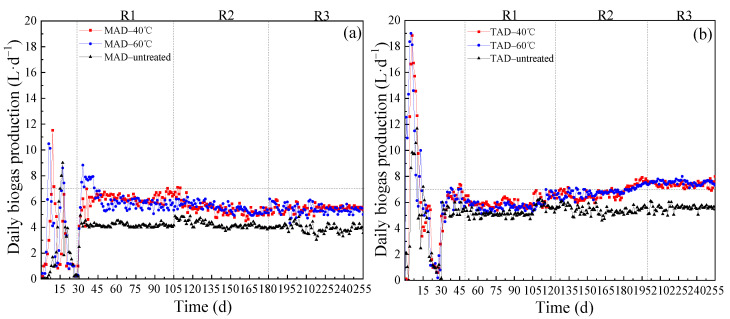
(**a**,**b**). Daily biogas production of CS under different pretreatment and AD conditions. (**a**): different pretreatment conditions for MAD; (**b**): different pretreatment conditions for TAD.

**Figure 2 ijerph-19-08027-f002:**
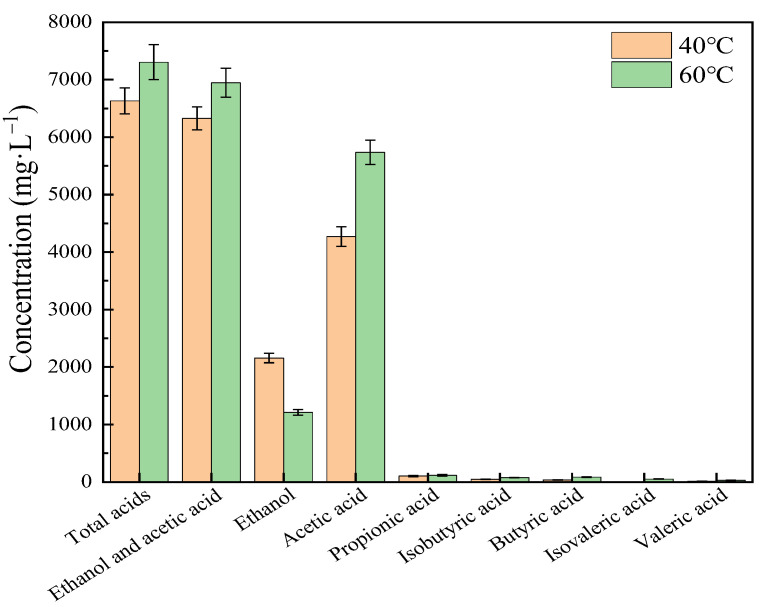
Individual VFAs and Ethanol concentrations of CS pretreated by 40 °C and 60 °C conditions.

**Figure 3 ijerph-19-08027-f003:**
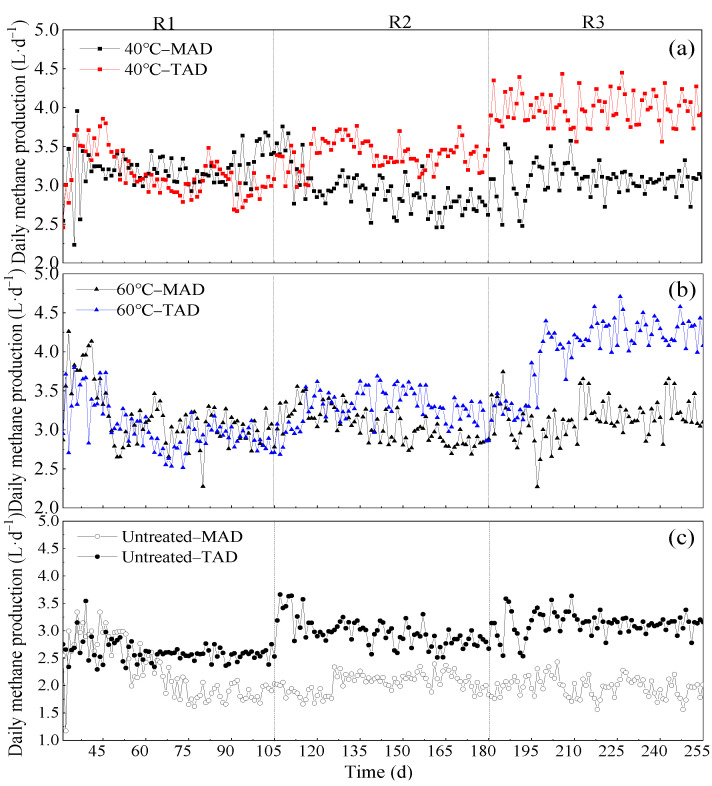
(**a**–**c**). Daily methane production of different pretreatment and anaerobic digestion temperatures. (**a**): 40 °C group; (**b**): 60 °C group; (**c**): untreated group.

**Figure 4 ijerph-19-08027-f004:**
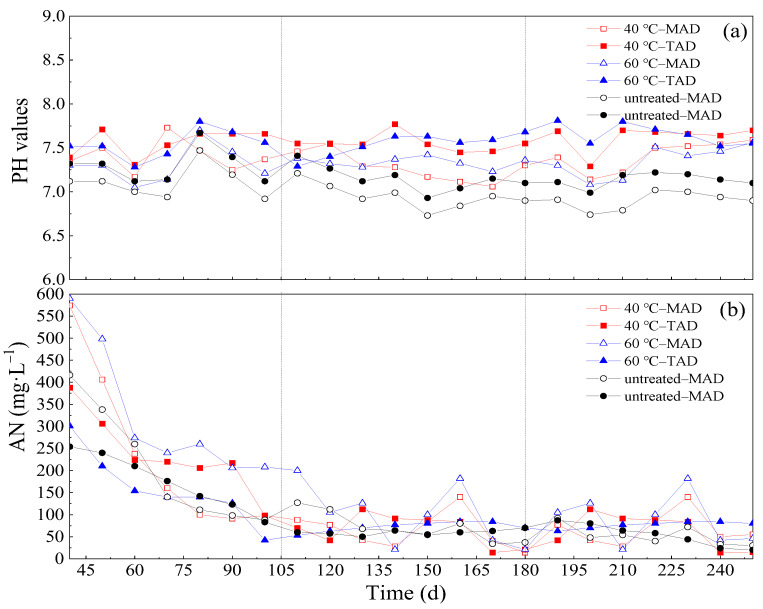
(**a**,**b**). The pH values and AN concentration of different pretreatment and anaerobic digestion temperatures. (**a**): pH; (**b**): AN.

**Figure 5 ijerph-19-08027-f005:**
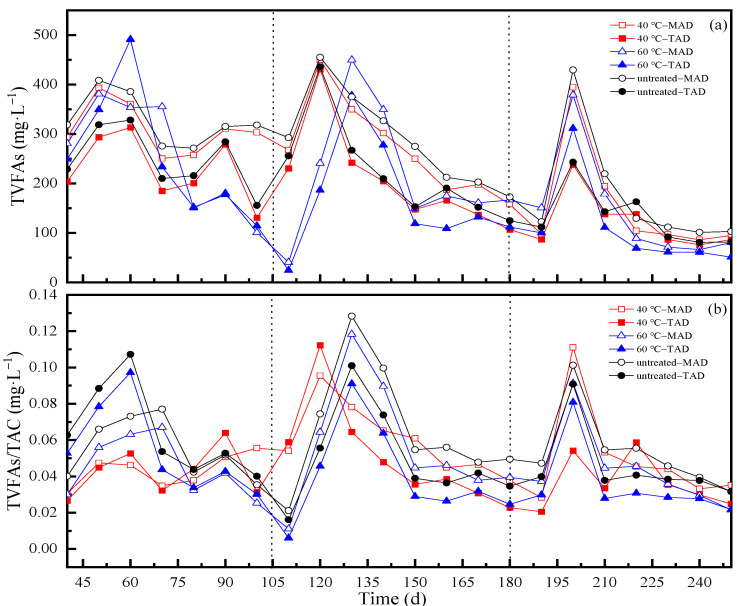
(**a**,**b**). The TVFAs and TVFAs/TAC of different pretreatment and anaerobic digestion temperatures. (**a**): TVFAs; (**b**): TVFAs/TAC.

**Table 1 ijerph-19-08027-t001:** Characteristics of corn stalk and inoculum before incubation *.

Parameter	Value (%)	
	Corn Stalk	Inoculum
TS (%) ^a^	91.05 ± 0.23	8.25 ± 0.03
VS (%) ^a^	87.80 ± 0.15	4.18 ± 0.05
TC (%) ^b^	43.42 ± 0.54	27.05 ± 0.25
TN (%) ^b^	0.68 ± 0.08	2.3 ± 0.53
C/N(%) ^b^	63.85 ± 0.56	11.76 ± 0.89
MLSS (g L^−1^) ^a^	ND	109.00 ± 2.18
Cellulose (%) ^b^	40.36 ± 0.60	ND
Hemicellulose (%) ^b^	20.07 ± 0.03	ND
Lignin (%) ^b^	11.41 ± 0.13	ND
LCH (%) ^b^	71.84 ± 0.76	ND

* Values are means ± SD (*n* = 3); ^a^ Content of fresh matter; ^b^ Content of dry matter; ND: Not determined; TS: Total Solids; VS: Volatile Solids; VSTC: Total Carbon; TN: Total Nitrogen; C/N: The ratio of Carbon to Nitrogen; MLSS: Mixed Liquid Suspended Solids; LCH: The addition of Lignin, Cellulose and Hemicellulose.

**Table 2 ijerph-19-08027-t002:** Comparisons of different anaerobic digestion performance *.

Pretreatment Temperature (°C)	AD Temperature (°C)	OLR (g·L^−1^·d^−1^)			OLR (g·L^−1^·d^−1^)			OLR (g·L^−1^·d^−1^)		
		1.6	1.8	2.0	1.6	1.8	2.0	1.6	1.8	2.0
		Biogas Yield (mL·g_vs_^−1^)			Methane Yield (mL·g_vs_^−1^)			Average Methane Content (%)		
40	35	524 ± 30	409 ± 20	319 ± 16	275 ± 20	220 ± 20	181 ± 16	52.49 ± 0.79	53.67 ± 0.95	56.90 ± 1.31
	55	513 ± 29	488 ± 27	450 ± 24	270 ± 19	255 ± 17	236 ± 14	52.46 ± 0.74	52.43 ± 0.62	52.55 ± 0.80
60	35	528 ± 31	417 ± 22	320 ± 16	280 ± 21	229 ± 12	184 ± 9	53.05 ± 0.86	55.01 ± 1.08	57.52 ± 1.52
	55	495 ± 28	478 ± 26	449 ± 24	258 ± 18	247 ± 16	236 ± 14	52.17 ± 0.59	51.66 ± 0.51	52.45 ± 0.71
untreated	35	383 ± 19	299 ± 17	230 ± 14	194 ± 11	153 ± 5	118 ± 16	50.60 ± 0.46	51.17 ± 0.58	51.30 ± 0.52
	55	444 ± 23	417 ± 22	340 ± 18	223 ± 11	213 ± 10	180 ± 9	50.22 ± 0.44	51.08 ± 0.41	52.94 ± 0.73

* Values are means ± SD (*n* = 3).

**Table 3 ijerph-19-08027-t003:** Comparisons of TVFAs under different pretreatment conditions *.

Pretreatment Temperature (°C)	Ethanol and Acetic Acid (mg·L^−1^)	Individual Acid Concentration (mg·L^−1^)							
		Ethanol	Acetic Acid	Propionic Acid	Isobutyric Acid	Butyric Acid	Isovaleric Acid	Valeric Acid	Total Acid
40	6326 ± 30	2156 ± 13	4270 ± 27	103 ± 10	47 ± 13	35 ± 3	4 ± 0.2	11 ± 0.6	6630 ± 32
60	6944 ± 41	1210 ± 9	5734 ± 12	115 ± 4	76 ± 4	86 ± 5	50 ± 3.6	30 ± 1.1	8303 ± 45

* Values are means ± SD (*n* = 3).

**Table 4 ijerph-19-08027-t004:** Changes of main compositions after pretreatment *.

Pretreatment Temperature (°C)	Cellulose	Hemicellulose	Lignin	LCH
40	36.5 ± 0.4	13.6 ± 0.1	10.6 ± 0.1	56.4 ± 0.3
60	35.4 ± 0.3	13.1 ± 0.1	10.5 ± 0.1	55.6 ± 0.2
untreated	40.3 ± 0.6	20.0 ± 0.2	11.4 ± 0.2	71.8 ± 0.7

* Values are means ± SD (*n* = 3).

**Table 5 ijerph-19-08027-t005:** Main components conversion rate of CS at different pretreatment and AD temperatures.

Pretreatment Temperature (°C)	AD Temperature (°C)	OLR (g·L^−1^·d^−1^)	Conversion Rate (%) *				
			Cellulose	Hemicellulose	Lignin	TS	VS
40	35	1.6	64.3 ± 1.2	60.1 ± 1.6	25.1 ± 0.7	50.0 ± 2.2	61.2 ± 1.7
		1.8	60.4 ± 1.6	56.7 ± 3.2	24.7 ± 0.6	48.7 ± 2.5	56.9 ± 2.6
		2.0	53.5 ± 3.3	49.2 ± 3.0	21.2 ± 0.4	46.3 ± 3.1	53.5 ± 2.5
	55	1.6	64.3 ± 3.5	60.2 ± 3.4	26.2 ± 0.8	50.6 ± 1.5	60.7 ± 2.8
		1.8	62.7 ± 2.8	58.6 ± 2.1	25.6 ± 0.7	48.9 ± 2.7	58.3 ± 1.5
		2.0	59.5 ± 3.7	55.4 ± 3.2	21.4 ± 0.4	47.1 ± 2.4	54.4 ± 3.0
60	35	1.6	64.9 ± 2.1	60.4 ± 3.1	25.4 ± 0.7	50.2 ± 1.8	61.9 ± 2.6
		1.8	59.3 ± 2.4	55.6 ± 4.2	21.6 ± 0.4	48.0 ± 2.2	55.7 ± 1.9
		2.0	53.5 ± 4.1	49.8 ± 3.4	19.8 ± 0.3	46.3 ± 2.4	54.2 ± 1.7
	55	1.6	61.7 ± 3.2	57.3 ± 2.2	25.3 ± 0.9	49.6 ± 2.6	57.7 ± 2.4
		1.8	60.2 ± 3.6	56.9 ± 3.0	24.9 ± 0.7	48.2 ± 1.7	56.0 ± 1.6
		2.0	59.4 ± 3.8	55.4 ± 2.1	20.4 ± 0.3	47.1 ± 2.3	55.1 ± 1.8
untreated	35	1.6	55.6 ± 2.0	53.2 ± 2.1	18.4 ± 0.2	46.1 ± 1.7	57.4 ± 2.6
		1.8	52.5 ± 2.2	51.6 ± 2.2	17.7 ± 0.2	45.2 ± 1.5	55.7 ± 1.8
		2.0	47.5 ± 2.1	45.5 ± 1.4	15.6 ± 0.1	43.3 ± 1.4	53.5 ± 1.5
	55	1.6	59.3 ± 3.3	55.4 ± 2.2	19.4 ± 0.2	47.5 ± 2.2	58.7 ± 2.3
		1.8	58.1 ± 3.1	52.7 ± 2.0	18.9 ± 0.2	46.8 ± 1.7	56.2 ± 1.9
		2.0	56.4 ± 3.0	49.4 ± 1.5	16.3 ± 0.1	44.9 ± 1.3	53.6 ± 1.6

* Values are means ± SD (*n* = 3).

**Table 6 ijerph-19-08027-t006:** Energy balance of MAD and TAD systems during the steady period.

Process	Item	MAD	TAD
Anaerobic digestion	E_o_ (kWh/mg VS)	3855.4	4012.6
Pretreatment	E_i,h_ (kWh/mg VS)	0 (untreated)	0 (untreated)
		298.67 (40 °C)	298.67 (40 °C)
		597.34 (60 °C)	597.34 (60 °C)
Mixing	E_i,m_ (kWh/mg VS)	108.0	276.3
Pumping	E_i,p_ (kWh/mg VS)	21.6	43.9
Anaerobic digestion	R_o/i_	29.7 (untreated)	12.5 (untreated)
		9.0 (40 °C)	6.48 (40 °C)
		5.3 (60 °C)	4.4 (60 °C)
Anaerobic digestion	ΔE (kWh/mg VS)	3725.8 (untreated)	3692.4 (untreated)
		3427.13 (40 °C)	3393.73 (40 °C)

E_o_: Energy input; E_i,h_: Energy for heating; E_i,m_: Energy for mixing; E_i,p_: Energy for pumping; R_o/i_: Ratio of E_o_ to E_i_; ΔE: Energy recovery.

**Table 7 ijerph-19-08027-t007:** Economic analysis of different groups.

Pretreatment Temperature (°C)	AD Temperature (°C)	Amount (Ton)	Reagent Cost (USD)	Water (USD)	Electricity Cost (USD)	Staff Cost (USD)	Sum (USD)	Methane Yield (m^3^·t^−1^)	Unit Methane Production Cost (USD·m^3^·CH_4_^−1^)
40	35	0.02	3.56	0.4266	3.5936	7.485	21.0852	181	0.0833
	55	0.02	3.56	0.4266	5.6603	7.485	23.1519	236	0.0726
60	35	0.02	3.56	0.4266	4.4762	7.485	21.9678	184	0.0867
	55	0.02	3.56	0.4266	6.9981	7.485	24.4897	236	0.0783
untreated	35	0	0	0.2558	3.3455	7.485	11.0863	118	0.1241
	55	0	0	0.2558	4.6282	7.485	12.369	180	0.0884

## Data Availability

Not applicable.

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
