# Peer review of "Determining Optimal Temperature Combination for Effective Pretreatment and Anaerobic Digestion of Corn Stalk"

_ijerph, 2022, doi:10.3390/ijerph19138027_

Round 1
Reviewer 1 Report
Article "Determining optimal temperature combination for effective
pretreatment and anaerobic digestion of corn stalk" is written precisely and results demonstrated in organized manner. References can be updated add more new references. Abstract and conclusion can be improved.
Reviewer 2 Report
Overall suggestions:
Please have your manuscript professionally proofread. Also, please add a nomenclature page for all abbreviations used before the introduction section. It is really easy to get lost with all the abbreviations, so the nomenclature page can be used as a reference when needed.
Make sure your figures are in-line with the text, not wrapping the text. Also, please add the inside borders to your tables because all the texts in the tables are nesting together. You can even increase the space between the rows.
* Introduction
- The unit in the sentence where you cited reference 13 has a different unit from references 5 and 6. Does the reference 13 really have higher TAD when you make the units consistent?
- What is the usual length of operation in other studies? Is a total of ~240 days considered long-term compared to literature and real-world applications? Please elaborate further with more explanations and citations as your main objective is to evaluate the long-term effect of biogas and methane production under different pre- and AD treatment conditions.
- Overall, please extend the introduction section to make your case better.
* Material and methods:
- Feedstock and inoculum: Why incubation times and respective inoculum times are different? Why did you incubate for 1 week at 35 C and 2 weeks at 55 C? How does this affect the experimental results?
- Table 1: Are the inoculum characteristics valid for before or after the incubation period as you mentioned seeding sludge in the text above table 1?
* Results and Discussion:
- Change your figure 2 from 2D bars to 2D columns. It would be easier to read and understand your data that way.
- Section 3.4: You mentioned that lignin degradation could release more cellulose and hemicellulose, but cellulose and hemicellulose contents for 40 and 60 C are still lower than the untreated contents even when the lignin is degraded. Please explain it.
- Section 3.5.1: How did you compare the effects of pH on the stability of the AD system when you didn't run any experiments at different pH levels?
- Section 3.5.2. In figure 4b, it is shown that the AN gradually reduced to 250 mg/L at the end of the first OLR (not 200 mg/L as you stated in the manuscript). Please explain how this changes the results compared to reference 35.
- Figure 4C is the most confusing figure. I suggest you make two separate figures for TVFAs and TVFAs/TAC.
- You explained that the time of TVFAs accumulation of 40C samples was 10 days faster and had more TVFAs than 60 C samples. I might agree with the accumulation amount for day 120, but I don't think you can say it is necessarily 10 days faster. There are times that 60 C TAD has higher TVFAs than 40C TAD. Also, you didn't explain what these results mean for 40C vs untreated and 60C vs untreated. Can you please justify and elaborate on your results? Additionally, please explain why TVFAs would be still low after R2 feeding time for about 30 days? What causes this slow increase? (e.g. 60C MAD and TAD between days 105 and 135).
- Please make sure the units on the figures and text are consistent with each other.
- Figure 4c TVFAs/TAC part: There are two red lines with square markers. One of them should be black. Also, what does "operating condition in each CSTR was at optimum level" mean in terms of 40C, 60C, and untreated? How do you decide which condition is better if all are optimum?
- Section 3.6.: How Ei,h was the same for 40C and 60C, and why?
* Conclusion
-If the energy consumption for heating to 40C and 60C is the same, no significant difference in enhancing methane yield between 40C and 60C, and all CSTR operating conditions are optimum according to Figure 4C, how can you conclude as the "optimum pretreatment temperature of CS was 40C for both MAD and TAD"?
It also seems like the MAD and TAD temperature is way more important than the pretreatment. Also, you almost gave nothing about the comparison of all parameters with the control. Please justify your conclusion with more details.
Reviewer 3 Report
Review (recommended major revision)
In manuscript authors describe the relationship between the biogas production process, as well as the temperature of pretreatment, as well as anaerobic digestion mechanisms. Comments, as well as suggestions are written in attachment. This should be duly considered.
In manuscript, titled “Determining optimal temperature combination for effective pretreatment and anaerobic digestion of corn stalk”, authors describe the relationship between the biogas production process, as well as the temperature of pretreatment, as well as anaerobic digestion mechanisms. Comments, as well as suggestions are written below.
Abstract
- “Pretreatment temperature is one of the decisive factors while using chemical regents to break the complex structures of lignocellulosic compounds (lignin, cellulose, and hem-icellulose (LCH))[2].” – Authors should tell a bit more about pretreatment itself, looking at some new works, assessing it. 10.21203/rs.3.rs-264148/v1, but others can be considered as well, improving review. This is presently slightly a drawback.
- The abbreviation MAD should be explained the first time it is used.
Introduction
- The current barriers in wider biogas usage should be mentioned as well.
- A quick overview of other pretreatment methods, such as DES, should be presented and compared to NaOH.
- Can the authors please comment on practicality of NaOH use for pretreatment of biomass, considering environmental impact and energy usage for production of NaOH.
- Add ‘’are’’ to the sentence ‘’Due to the natures of microorganisms which facilitate anaerobic processes in reactors, bioconversions commonly carried out under either mesophilic (30–45°C) or thermophilic (50–65°C) conditions’’.
- Did the authors mean to write ‘’Among the biochemical stages in anaerobic process, methanogenesis performs effectively between 35 °C and 55 °C reactor temperatures’’.
- Please correct ‘’…has some advantages over mesophilic anaerobic digestion MAD…’’ to only the abbreviation to be used.
Materials and methods
- Please specify cold and dry place. Is this room temperature, a fridge?
- Why was the innoculum incubated for one week at 35 °C and two weeks at 55 °C?
Results and discussion
- In Figure 1 it is not clear what the methane content is. Green points and line are barely visible, change of color is suggested.
- Please revise the sentence ‘’ During MAD processes, the average methane content showed respective increment from 52.49% to 56.90% and from 53.05% to 57.52% …’’ as it is unclear what the basis for percentage calculation is.
- It seems that the peak of the production at MAD-untreated comes with a delay. What is the reason for this?
- Can the authors please explain the drop of biogas production to almost 0 at 30 days?
- Table 2: What is the role of the bolded text? It should be explained in the table caption. The table is not easily readable.
- Figure 3: The same color of the lines for different experiments are confusing and difficult to read. Furthermore, is it necessary to connect the points with lines as this is not a continuous process but represents each day separately? Same comment applies to Fig 1.
- Is it necessary to introduce abbreviation DMP, considering it is never used?
- Is in energy balance of pretreatment considered only heat? How about production of NaOH?
- In Table 6 it is unclear what the results represent, please denote what are Eo, Ei,h etc. It seems that the highest ΔE is obtained without pretreatment, how would the authors justify pretreating the samples and using more energy for additional steps?
Round 2
Reviewer 2 Report
It seems the authors made extensive changes based on my suggestion. Thank you for that. There are a couple of comments that I suggest. Please find them in the attached PDF file. My comments are in green.

Author Response
请参阅附件。

Reviewer 3 Report
Review (recommended major revisions)
Comments were not addressed notably - the review of literature is not up to date.
Author Response
请参阅附件。
